# Delays in seeking healthcare and its determinants among malaria patients in Ethiopia: A systematic review and meta-analysis

**Moges Tadesse Abebe**[1]*, **Tesfahun Zemene Tafere**[2], **Kaleab Tesfaye Tegegne**[3], **Dessie Alemnew Shiferaw**[4], **Yosef Aragaw Gonete**[5], **Tadele Kassahun Wudu**[6], **Getnet Azanaw Takele**[7], **Muluken Chanie Agimas**[8]

**1** Department of Pediatric and Child Health Nursing, College of Health Science, Debark University, Debark, Ethiopia, **2** Department of Health Systems and Policy, Institute of Public Health, College of Medicine and Health Sciences, University of Gondar, Gondar, Ethiopia, **3** Department of Public Health, College of Health Science, Debark University, Debark, Ethiopia, **4** Department of Nursing, College of Health Science, Debark University, Debark, Ethiopia, **5** Department of Midwifery, College of Health Science, Debark University, Debark, Ethiopia, **6** Department of Statistics, College of Natural and Computational Science, Debark University, Debark, Ethiopia, **7** Department of Midwifery, College of Medicine and Health Sciences, University of Gondar, Gondar, Ethiopia, **8** Department of Epidemiology and Biostatistics, Institute of Public Health, College of Medicine and Health Sciences, University of Gondar, Gondar, Ethiopia

* moges7045@gmail.com

## Abstract

Delays in seeking healthcare among malaria patients are typically associated with an increased risk of severe disease and mortality. Determining the pooled prevalence and determinants of delays in seeking healthcare may help reduce morbidity and mortality. Therefore, the aim of this systematic review and meta-analysis was to determine the pooled prevalence of delays in seeking healthcare and its determinants among malaria patients in Ethiopia.

### Methods

PubMed, Cochrane Library, Scopus, Web of Science, Google Scholar and Google were searched. Cross-sectional and case–control studies about delays in seeking healthcare for more than 24 hours were included. STATA 17 was used to analyze the data. Heterogeneity across studies was assessed via the $I^2$ test. A funnel plot and Egger's test were used to assess publication bias. Subgroup analyses were performed by region and study setting. Sensitivity analysis was performed to determine the influence of individual studies.

### Results

A total of 18 articles with 7371 participants were included in this review. The pooled prevalence of delay in seeking healthcare was 67% (95% CI: 51%-84%). Age less than 15 years (OR: 2.27, 95% CI: 1.34-3.85), inability to read and write (OR: 3.36, 95% CI: 1.18-9.69),

**Data availability statement:** Data are available with the paper.

**Funding:** The author(s) received no specific funding for this work.

**Competing interests:** I have read the journal's policy and the authors of this manuscript have no competing interests.

**Abbreviations:** CBHI: community-based health insurance, CI: confidence interval, JBI: Joanna Briggs Institute, OR: odds ratio, WHO: World Health Organization.

travel to health institutions on foot (OR: 2.77, 95% CI: 1.71-4.49), and by horse (OR: 2.76, 95% CI: 1.57-4.84), living far from a health institution (OR: 2.65, CI: 1.37-5.13), not having a family history of death (OR: 3.04, 95% CI: 2.14-4.33), and not being a member of community-based health insurance (OR: 7.14, 95% CI: 1.09-46.63) were significant determinants of delays in seeking healthcare.

## Conclusion

The pooled prevalence of delays in seeking healthcare was high, and most of the determinants were modifiable. These findings underscore the need for targeted interventions to address these barriers and improve timely access to healthcare for affected populations.

## Introduction

Malaria is a protozoon of the genus *Plasmodium*, namely, *Plasmodium falciparum*, *Plasmodium malariae*, *Plasmodium ovale*, *Plasmodium vivax*, and *Plasmodium knowlesi* [1,2]. The parasite is transmitted to human via the bite of female Anopheles mosquitoes. The clinical feature of malaria, which differentiates it from other infectious diseases, is febrile paroxysms alternating with relative wellness. These febrile paroxysms include high fever, rigors, sweating, and headache [2,3].

In 2022, the global malaria prevalence was 249 million, where 94% of which were from Africa and more than six hundred people died worldwide, with more than 50% of those deaths occurring in African countries. Malaria remains one of the top ten causes of morbidity and mortality worldwide [4,5].

Approximately 75% of Ethiopians are at risk of malaria infection. The prevalence of malaria in Ethiopia varies by geographic district. Depending on the nature of the epidemic in each district, the prevalence varies from as low as 5% to as high as 50% [6–12].

The healthcare-seeking behavior of the community plays an important role in effectively managing of malaria [13]. The World Health Organization (WHO) recommends that malaria should be treated within 24 hours of the onset of symptoms to prevent severe malaria related disease and mortality. A systematic review conducted on delays in seeking healthcare among malaria patients also indicated that the probability of severe malaria was greater among patients who sought care more than 24 hours after malaria symptom onset than among those who sought care less than 24 hours after malaria symptom onset [14]. Research findings have also indicated that early diagnosis of malaria reduces complications and mortality among malaria patients [15, 16]. Severe and cerebral malaria are among the commonest complications of malaria, which could be attributed to an avoidable delay in seeking healthcare [17].

However, studies conducted in several regions of Ethiopia have shown that few patients actually seek medical attention within the recommended 24 hours after the onset of symptoms [18–27]. The results of studies conducted in Ethiopia have also shown that the prevalence of delays in seeking healthcare among malaria patients after 24 hours of symptom onset varies widely [18,23,25,28,29].

Among studies conducted in Ethiopia, several factors influenced the delay in seeking healthcare among malaria patients. These included sociodemographic factors such as age, sex, marital status, educational status, occupational status and income [30–32]. Other determinants include death in the family, side effects of antimalarial drugs, previous malaria infection, behavioral factors such as khat chewing and alcohol consumption, distance and transport-related factors, and self-medication practices that account for delays in seeking healthcare

among malaria patients [33–37]. There are also inconsistent findings among different studies about the associations among these determinants. For example, some studies reported that delays in seeking healthcare occurred when the household head decided to seek care [31,38], whereas another study reported that early seeking healthcare occurred when the household head decided to seek care [35].

Hence, this review pooled the findings from the inconsistent findings of the prevalence and determinants among primary studies conducted on delays in seeking healthcare and its determinants. This is the first systematic review and meta-analysis on the delay in seeking healthcare for malaria in Ethiopia. Determining the pooled prevalence of delays in seeking healthcare and their determinants among malaria patients in Ethiopia was the goal of this meta-analysis. The results could benefit healthcare leaders, policymakers and health professionals in designing strategies to reduce delays in seeking healthcare among malaria patients in Ethiopia.

## Methods

This systematic review and meta-analysis adhered to the Preferred Reporting Items for Systematic Reviews and Meta-Analyses (PRISMA 2020) guidelines [39] (S1 File).

### Search strategies

Before database searching, we assessed the prospective register of systematic reviews (PROSPERO) to check for duplicated work, and there was no registered review with our title. The online databases; PubMed, Cochrane Library, Scopus, Web of Science, Google Scholar and Google were searched from May 1, 2024 to June 15, 2024 for published articles. Unpublished studies were identified from Google and the institutional repositories of online universities, namely, Addis Ababa and Jima University. The references of identified studies were also searched for additional studies. We conducted a literature search via medical subject headings (MeSH). The search terms used were "malaria", "patient delay", "treatment seeking delay", "seeking healthcare delay", and the "Ethiopia". Boolean operators (AND and OR) were used to combine terms. ((Malaria[MeSH Terms]) AND (Patient delay OR treatment seeking delay OR seeking healthcare delay[MeSH Terms])) AND (Ethiopia) were used to search PubMed.

### Inclusion criteria

Cross-sectional and case-control studies conducted in Ethiopia on delays in seeking healthcare for malaria patients were included. Studies with a two-by-two table of factors associated with delays in seeking healthcare for malaria were also included. In addition, studies that used 24 hours of cutoff points to determine the presence or absence of delays in seeking healthcare for malaria, and published and gray literature conducted in English at any time were included.

### Exclusion criteria

Studies that were fully inaccessible excluded. We also excluded studies in which the outcome of interest was not met with the aim. Additionally, review articles and case reports were excluded.

### Outcome measurement

The primary outcome of this meta-analysis was the pooled prevalence of delays in seeking healthcare and the secondary outcome was determinants of delays in seeking healthcare among malaria patients in Ethiopia.

**Delay in seeking healthcare:** The time period from onset of malaria symptoms until first visit to health care facility. The patient was said to be delayed if s/he visited a health facility 24 hours after the onset of one of the malaria symptoms.

**Far distance:** If the patient travels a distance> 3k.m or takes > 30 minutes to reach the health facility.

**Symptoms:** any sensation or change in body function that is experienced by a patient.

**Onset of symptoms:** This is the time or day when the patient first becomes aware of the symptoms.

## Study selection

Following the database search, we exported the data to the EndNote 7 citation manager and eliminated any duplicates. Two reviewers (MT and GA) independently evaluated the titles and abstracts. Any differences between the two reviewers were addressed by involving a third reviewer (DA) and conducting a comprehensive review and discussion among the team members to reach an agreement.

## Data quality control measures

The Joanna Briggs Institute (JBI) critical appraisal checklist for cross-sectional and case-control studies was used to assess the quality of the included studies [40]. Studies with a score of 5 or more out of 8 criteria for cross-sectional studies and 7 or more out of 10 criteria for case-control studies were included. The quality of each study was independently evaluated by two authors (M. T., and M. C.). Discrepancies between the two were resolved by discussion with a third independent reviewer (K.T.) (S2 File).

## Data extraction

A standardized Excel data extraction spreadsheet was used to extract the data. The name of the authors, year of study, region, sample size, frequency of delay and odds ratio with CIs for the determinant factors were extracted. Two independent reviewers (MT and TK) extracted the data and cross-checked their consistency (S3 File).

## Analysis

Statistical analyses were performed by using STATA version 17. The heterogeneity test was conducted via I squared ($I^2$) statistics. The fixed effects model was used when the $I^2$ value was less than 50%, whereas the random effects model was used when the $I^2$ value was greater than 50%. The pooled prevalence of delay in seeking healthcare among malaria patients was determined via a random-effects method due to observed heterogeneity. To reduce potential random variations between studies; the sources of heterogeneity were analyzed using subgroup analysis. Publication bias was checked by visual inspection of the funnel plots and Egger's weighted regression test at a p-value < 0.05. The asymmetry of the funnel plot was also used as an indication of publication bias. Furthermore, trim and fill analysis was conducted for the presence of publication bias during Egger's test. Sensitivity analysis was performed by removing each study at a time to identify the impact of each study on the overall effect size. We adhered to the PRISMA 2020 checklist thorough the manuscript to handle any missing data.

# Results

## Selection results

In this systematic review and meta-analysis, a total of 1171 articles were retrieved from online databases. Among them, 929 duplicated records were removed. Among 242 articles, 108 articles were

excluded by screening the titles and abstracts, where 92 did not met the outcome of interest and 16 were only abstracts. The remaining 134 full articles were assessed for eligibility, where 123 did not report the outcome of interest, 6 were case reports and 5 were review articles. Eighteen research articles were included in the final systematic and review meta-analysis [18–27,30–37] (Fig 1).

## Characteristics of the included studies

This systematic review and meta-analysis included 18 articles with a total of 7371 participants. Among them, eight articles used case-control designs [30–37] and ten articles used cross-sectional designs [18–27]. Four articles with 1272 children were also included [30,32,34,37]. All studies were conducted between 2000 and 2022. Seven of these studies were from the Oromia region, 6 were from the Amhara region, 1 was from the Tigray region, 1 was from the Southwest Ethiopia region, 1 was from the Central Ethiopia region, 1 was from the Gambela region and 1 was from Benshangul Gumz (Table 1).

## Prevalence of delays in seeking healthcare

Ten cross-sectional studies with 4212 participants were included in this meta-analysis. First, we conducted a meta-analysis with a fixed effects model to determine the degree of heterogeneity. However, the degree of heterogeneity was high, and we conducted the analysis

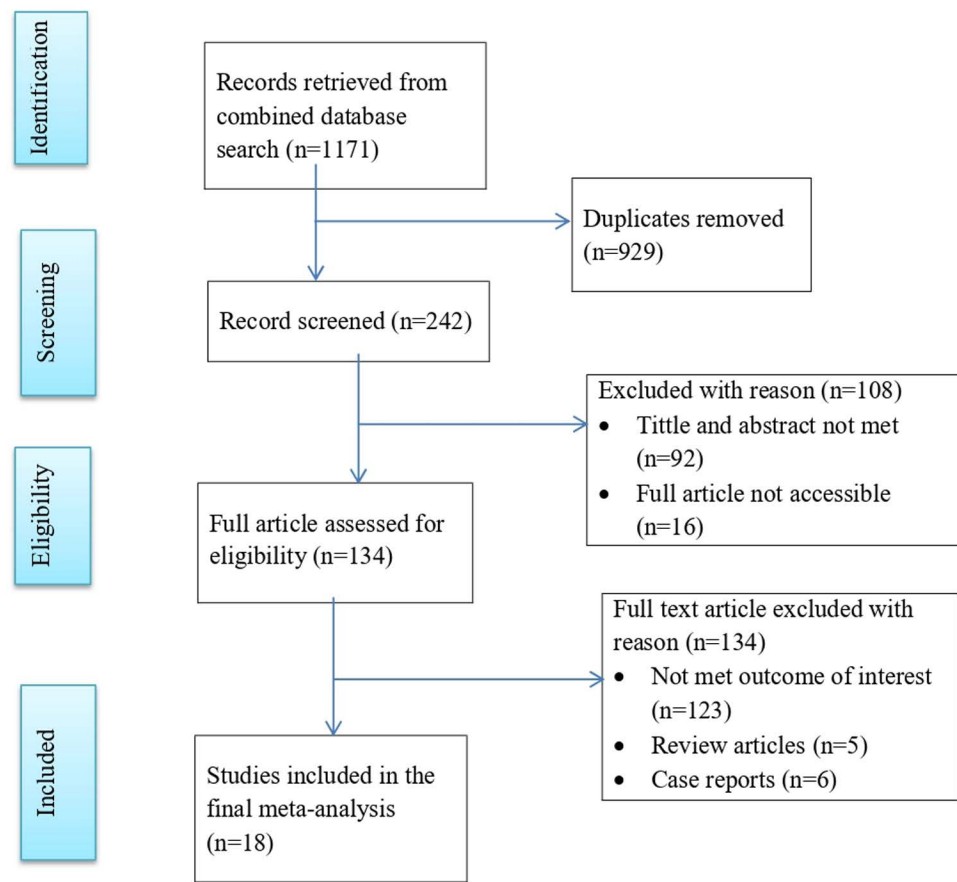

**Fig 1. PRISMA flow chart of study selection for systematic review and meta-analysis of delays in seeking healthcare among malaria patients in Ethiopia.**

**Table 1. Characteristics of the included studies on delays in seeking healthcare among malaria patients in Ethiopia.**

| s.n. | Author | Study year | Region | Study design | Sample size |
|---|---|---|---|---|---|
| 1 | Goshu and Tafasa [31] | 2022 | Oromia | Case control | 322 |
| 2 | Shiferaw, Geremew et al. [32] | 2016 | Southwest | Case control | 354 |
| 3 | Getahun, Deribe et al. [30] | 2010 | Oromia | Case control | 310 |
| 4 | Tiruneh, Gebrege et al. [36] | 2014 | Amhara | Case control | 318 |
| 5 | Shumerga, Hebo et al. [34] | 2017 | Gambela | Case control | 306 |
| 6 | Turuse, Gelaye et al. [37] | 2013 | Central | Case control | 302 |
| 7 | Regassa, Taffere et al. [26] | 2017 | Amhara | Cross-sectional | 394 |
| 8 | Tesfahunegn, Zenebe et al. [35] | 2019 | Tigray | Case control | 322 |
| 9 | Workineh and Mekonnen [27] | 2017 | Amhara | Cross sectional | 680 |
| 10 | Alga, Wasihun et al. [33] | 2022 | Amhara | Case control | 322 |
| 11 | Dida, Darega et al. [24] | 2014 | Oromia | Cross sectional | 297 |
| 12 | Mitiku and Assefa [25] | 2014 | Benshangul Gumz | Cross sectional | 302 |
| 13 | Deressa, Ali et al. [22] | 2003 | Oromia | Cross sectional | 481 |
| 14 | Deressa [21] | 2003 | Oromia | Cross sectional | 1155 |
| 15 | Deressa, Chibsa et al. [23] | 2000 | Oromia | Cross sectional | 392 |
| 16 | Birhanu, Abebe et al. [19] | 2014 | Oromia | Cross sectional | 116 |
| 17 | Belay, Gelana et al. [18] | 2018 | Amhara | Cross sectional | 246 |
| 18 | Dejazmach, Alemu et al. [20] | 2020 | Amhara | Cross sectional | 149 |

with the random effects model. Hence, the pooled prevalence of delay in seeking healthcare among malaria patients according to the random effects model was 0.67 (95% CI: 0.51, 0.84) $I^2$ = 99.6%, P < 0.001 (Fig 2).

## Subgroup analysis

Subgroup analysis based on region and study setting revealed that the Oromia region had the highest proportion of patients with delayed seeking of healthcare (0.84, 95% CI: 0.76-0.92), and the Amhara region had the lowest proportion of patients with delayed seeking of healthcare (0.44, 95% CI: 0.08-0.81) (Fig 3).

The proportion was also greater among community based cross-sectional studies (0.76, 95% CI: 0.63-0.88) than among institution based cross-sectional studies (0.62, 95% CI: 0.33-0.92) (Fig 4).

## Publication bias and heterogeneity

The presence of publication bias was tested by Egger's test and graphically by using a funnel plot. Visual inspection of the funnel plot revealed an asymmetric distribution (Fig 5). The Regression-based Egger test for small-study effects yielded a p-value of 0.002, indicating publication bias between studies. Therefore, we conducted nonparametric trim and fill analysis of publication bias. Hence, two studies were imputed and we found a proportion of 0.62 (95% CI: 0.48-0.77).

## Sensitivity analysis

We performed a leave-one-out meta-analysis by removing each study step by step to evaluate the effect of a single study on the overall effect estimate. The results indicated that the study conducted by Belay, Gelana et al. influenced the pooled effect size (Fig 6).

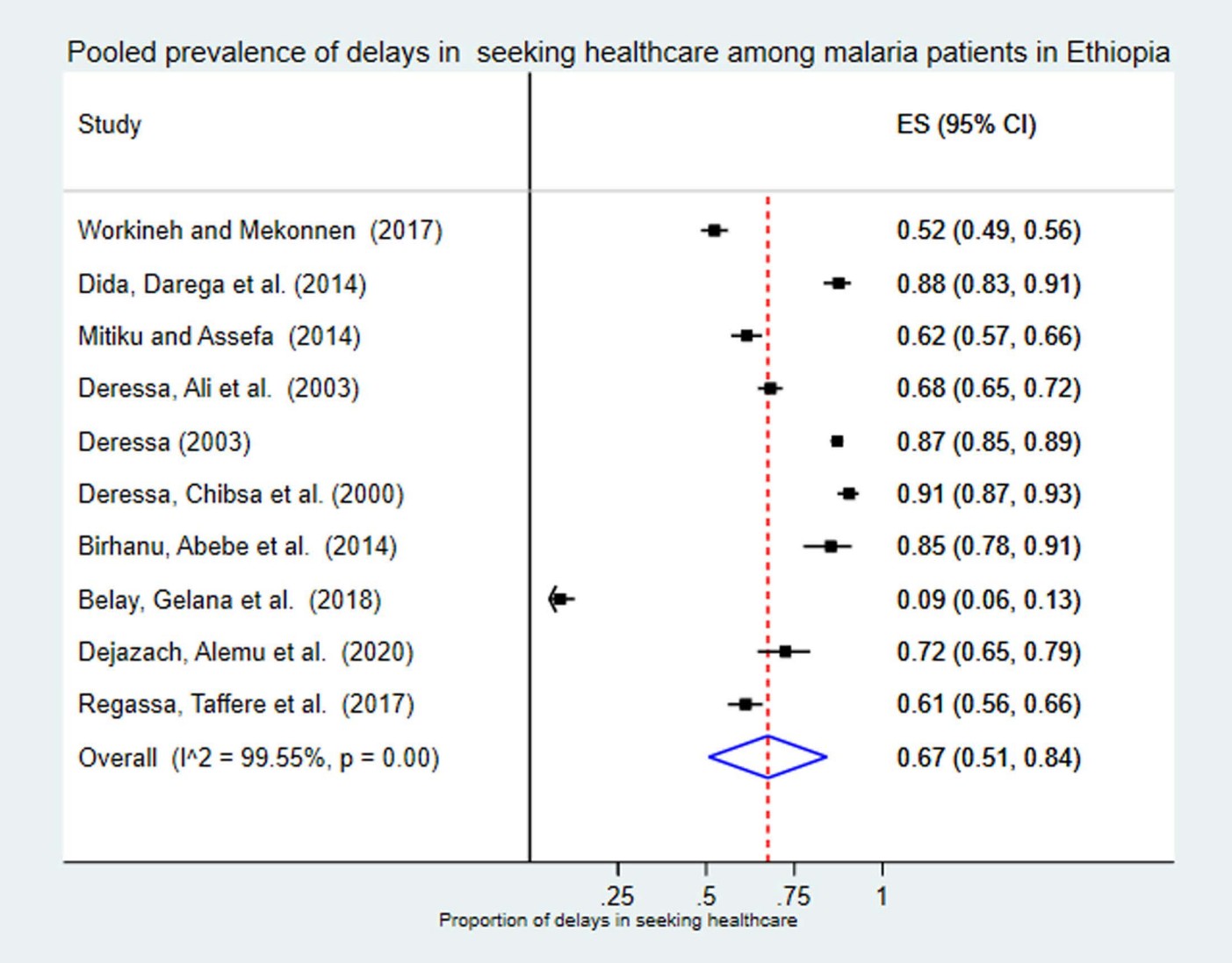

**Fig 2. Forest plot showing the pooled prevalence of delays in seeking healthcare among malaria patients in Ethiopia.**

## Determinants of delays in seeking healthcare

Through eleven studies with 3927 participants, more than 30 variables were subjected to a meta-analysis to identify factors associated with delays in seeking healthcare among malaria patients in Ethiopia. Among these patient-related variables, seven (patients less than 15 years old, patients who went to health institutions on foot, patients who went to health institutions by horse, patients who lived far from health institutions, patients who had no family history of death and patients who were not members of the CBHI) were significantly associated with delays in seeking healthcare among malaria patients. For the model, we used a fixed effects model when the $I^2$ value was less than 50%, and we used a random effects model when the $I^2$ value was greater than 50% (Table 2).

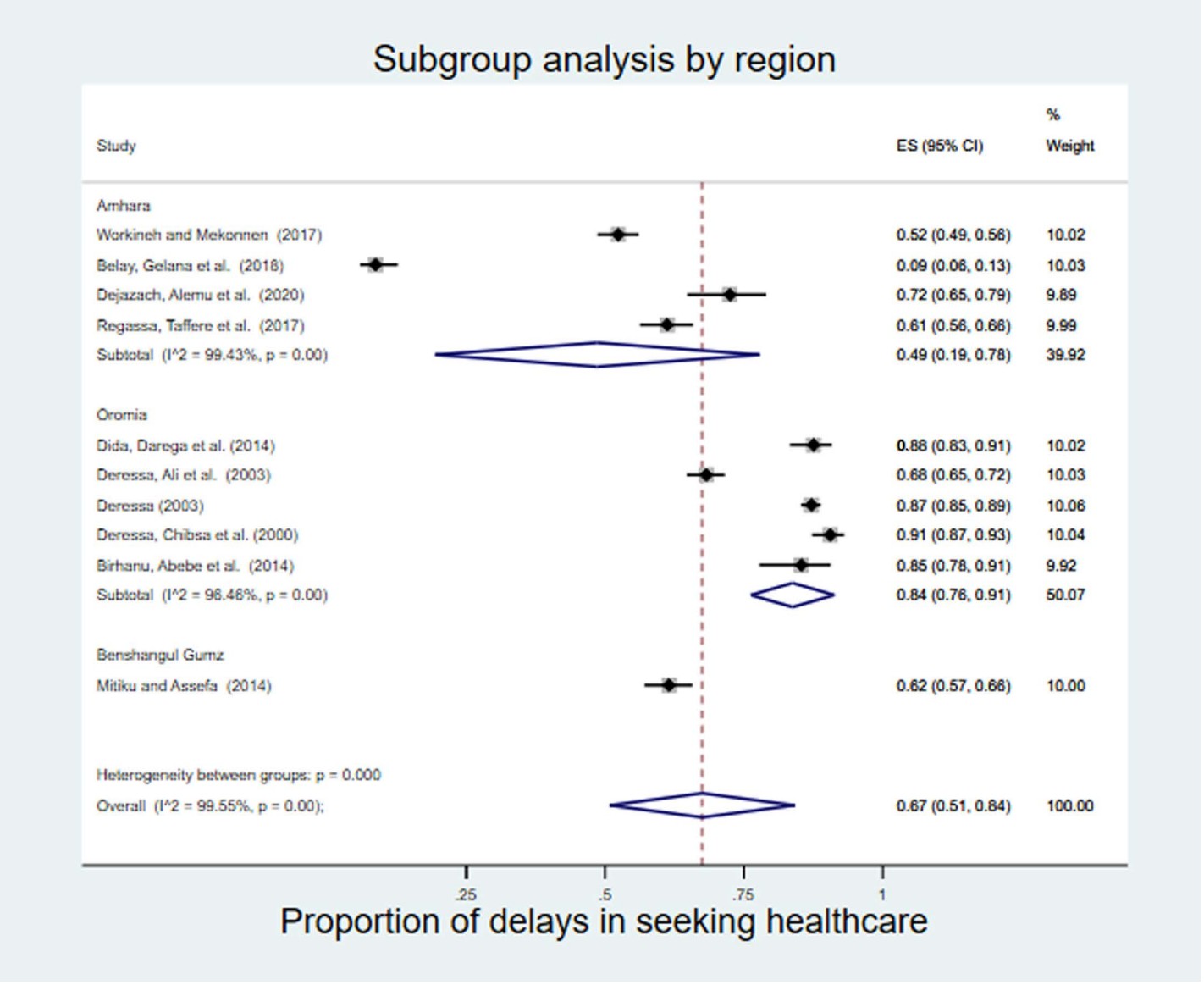

**Fig 3. Subgroup analysis by region of delay in seeking healthcare among malaria patients in Ethiopia.**

Age was one of the determinants of delays in seeking health care among malaria patients. Children younger than 15 years were 2.27 times more likely to be delayed than adults older than 29 years were (OR: 2.27, 95% CI: 1.34-3.85).

Patient literacy level was also one of the factors associated with delays in seeking healthcare among malaria patients. Patients who could not read and write were 3.36 times more likely to be delayed than those with a secondary education or above (OR: 3.36, 95% CI: 1.18-9.69).

The mode of transportation was the other determinant of seeking healthcare delay. Patients who went to health institutions on foot were 2.77 times more likely to be delayed than patients who were transported by car were (OR: 2.77, 95% CI: 1.71-4.49). In addition, patients who were transported by horse were 2.76 times more likely to be delayed than patients who were transported by cars were (OR: 2.76, 95% CI: 1.57-4.84).

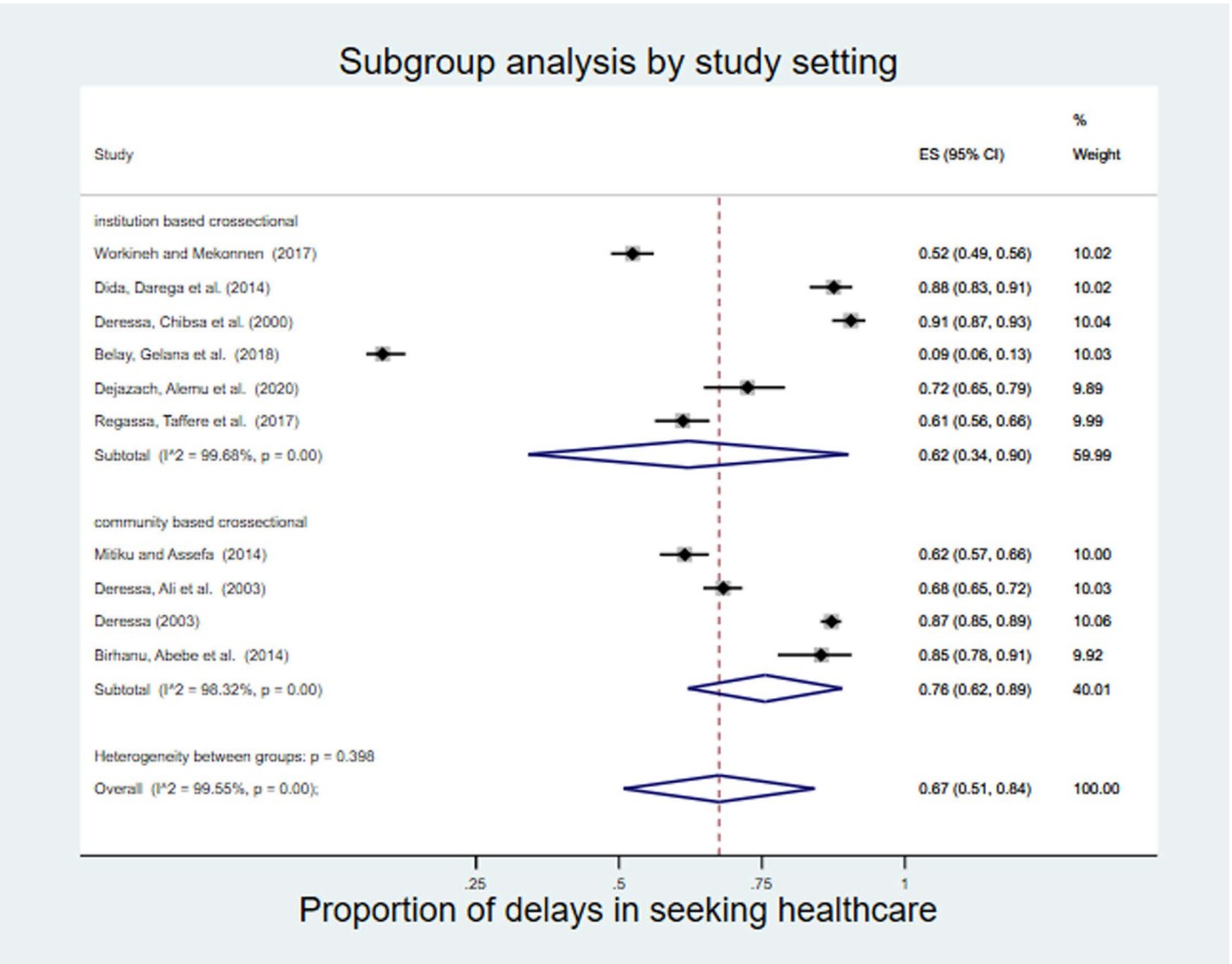

**Fig 4. Subgroup analysis by study setting on delays in seeking healthcare among malaria patients in Ethiopia.**

Patients who were far from health institutions were 2.65 times more likely to be delayed than patients who came from locations closer to health institutions were (OR: 2.65, 95% CI: 1.37-5.13).

No experience of death among family members of interviewed patients was also a determinant of delay. Patients who had no history of family death were 3.04 times more likely to be delayed than patients with a family history of death were (OR: 3.04, 95% CI: 2.14-4.33).

Not being a member of the CBHI was a significant determinant of delay. Patients who were not members of the CBHI were 7.14 times more likely to be delayed than those who were members of the CBHI (OR: 7.14, 95% CI: 1.09-46.63).

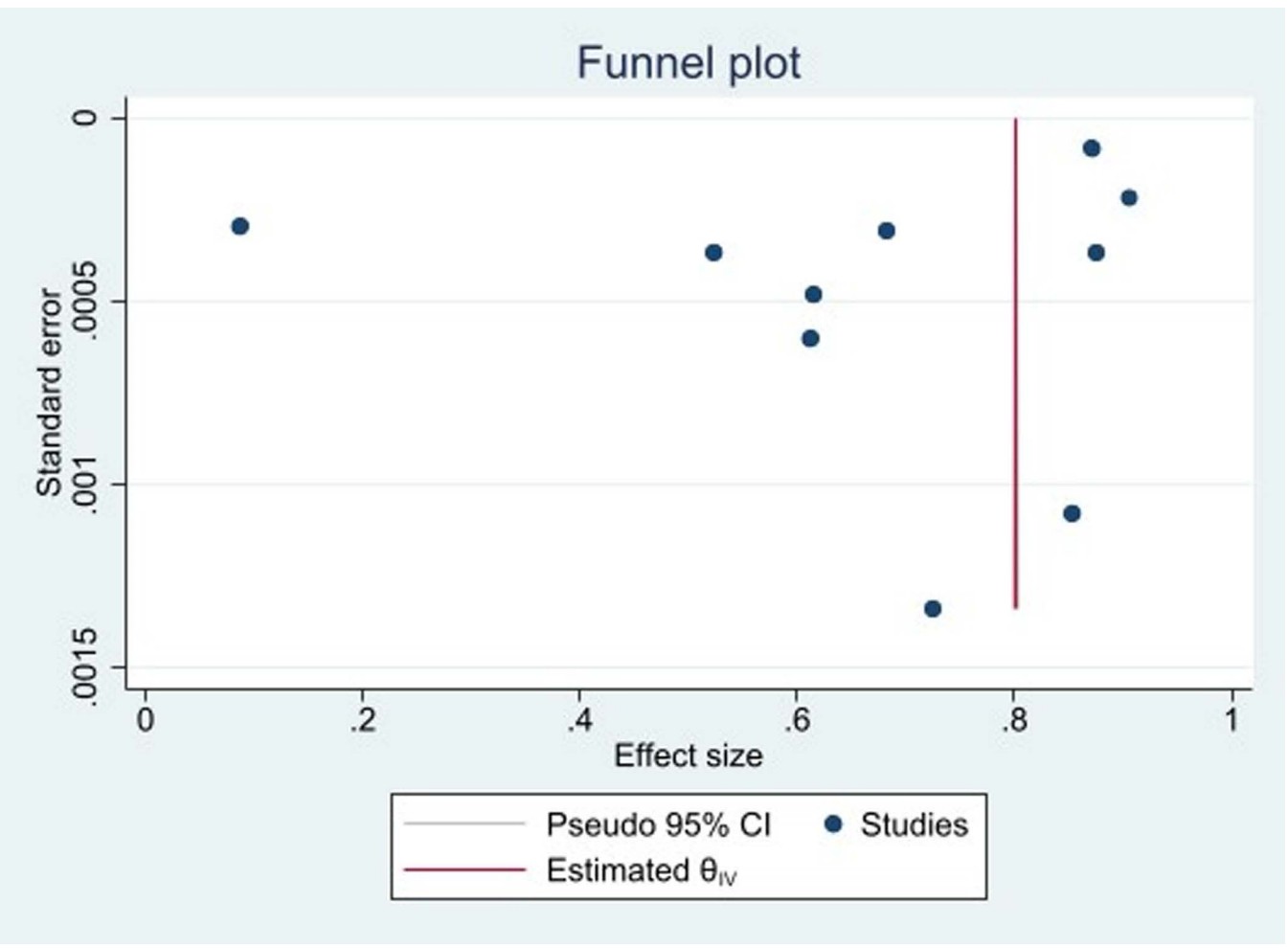

**Fig 5. Funnel plot showing publication bias among the included studies.**

## Discussion

This meta-analysis aimed to determine the pooled prevalence and determinants of delays in seeking healthcare among malaria patients in Ethiopia. The meta-analysis results revealed that the overall prevalence of patients with delayed seeking healthcare among malaria patients in Ethiopia was 67%. As a result, 67% of patients waited longer than the recommended 24 hours after the onset of malaria symptoms before visiting a health facility. This result was supported by studies conducted in Myanmar [41] and Tanzania [42], but it was lower than that of a study conducted in China, in which 85.6% of patients were delayed [43]. This reflects a failure of patients seeking healthcare in a timely manner, as expected, which may challenge malaria control programs to meet national and international goals. This discrepancy could be due to socioeconomic differences between these countries. To prevent malaria-related complications and death, researchers recommended treating malaria as soon as possible within 24 hours of symptom onset [43]. Early seeking healthcare behavior is also important for early malaria detection and treatment, thus reducing the transmission cycle [44].

Although the included studies did not represent all regions of the country, there were regional differences among the included studies that could be explained by the fact that the healthcare

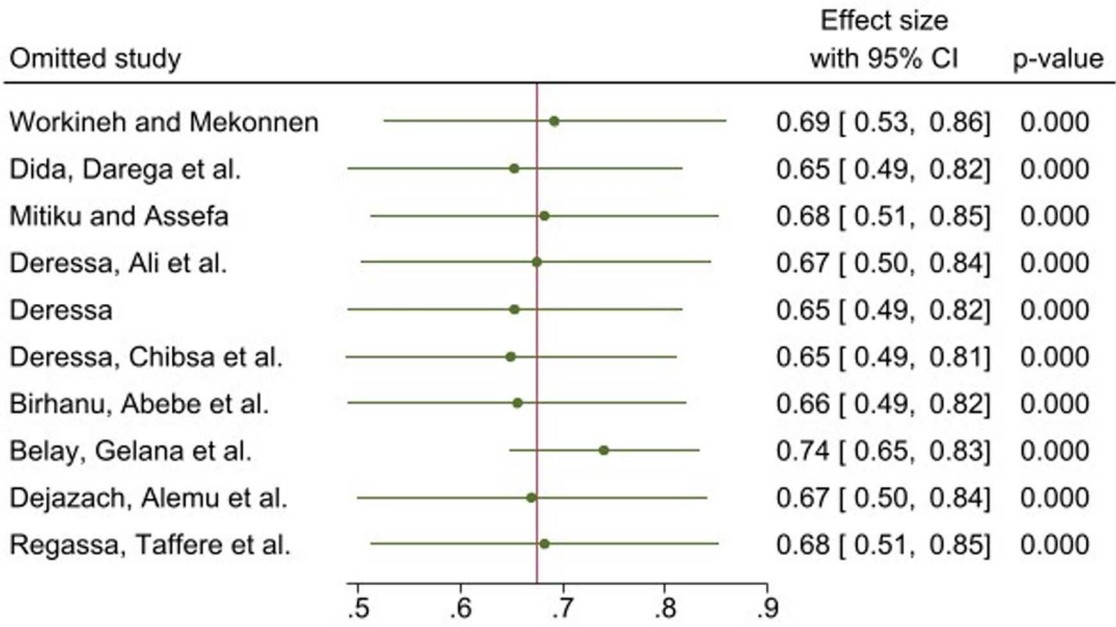

**Fig 6. Sensitivity analysis of the pooled prevalence for each study removed one at a time.**

**Table 2. Meta-analysis results of factors associated with delays in seeking healthcare among malaria patients in Ethiopia.**

| Determinants (references) | Comparison | Number of studies | Sample size | OR (95% CI) | P-Value | I² (%) | Heterogeneity test (p-value) | Model used |
|---|---|---|---|---|---|---|---|---|
| Age less than 15 years [31,33] | Age greater than 29 | 2 | 644 | 2.27 (1.34-3.85) | 0.002 | 0.00 | 0.88 | Fixed effect |
| Cannot read and write [24,33,35,37] | Secondary and above | 4 | 1243 | 3.36 (1.18-9.61) | 0.024 | 72.6 | 0.01 | Random effect |
| Patients who went to health institution on foot [26,33] | Patients who went to health institution by Car | 2 | 716 | 2.77 (1.71-4.49) | <0.001 | 0.00 | 0.46 | Fixed effect |
| Patients who went to health institution by horse [26,33] | Patients who went to health institution by Car | 2 | 716 | 2.76 (1.57-4.84) | <0.001 | 0.00 | 0.66 | Fixed effect |
| Far distance [24,26,27,30,32,36,37] | Near distance | 7 | 2655 | 2.65 (1.37-5.13) | 0.004 | 89 | <0.0001 | Random effect |
| Had no history of death [26,30,34,37] | Had history of death | 4 | 1312 | 3.04 (2.14-4.33) | <0.001 | 0.00 | 0.79 | Fixed effect |
| No member of CBHI [24,36] | Member of CBHI | 2 | 322 | 7.14 (1.09-46.6) | 0.04 | 96.2 | <0.001 | Random effect |

seeking behavior of the community may vary among different regions. Compared with institution-based cross-sectional studies, Community based cross-sectional studies had a greater proportion of delays in seeking healthcare than institution based studies, possibly because of the variation in the measurement tool used and the high rate of recall bias in community based studies.

Among the various variables analyzed; children younger than 15 years, patients who could not read and write, patients who went to health institutions on foot or by horse, patients who lived far from health institutions, patients who were not members of the CBHI, and patients who had no history of family death were factors associated with delays in seeking healthcare.

The meta-analysis revealed that the odds of delay in seeking healthcare among children aged less than 15 years were 2.27 times more likely to be delayed than those among adults aged greater than 29 years. Children may have difficulty understanding and expressing the symptoms and severity of the illness, causing parents to delay seeking medical attention until their children can recognize the illness and become seriously ill [45, 46]. This finding was also supported by a study conducted in Myanmar [41] and Nigeria [47], but it was contradicted by another study conducted in Myanmar where patients older than 30 years were delayed compared with younger patients [48]. Therefore, parents of children, especially in malaria-endemic areas, must receive special health education to seek healthcare early for children with symptoms. Additionally, patients who were unable to read and write were 3.36 times more likely to be delayed than those with a secondary education and above. This finding was supported by studies conducted in Myanmar [48], Indonesia [13] and Kenya [49]. This may be because as the level of education increases, awareness of the signs and symptoms of the disease and methods of prevention increase, leading to early healthcare seeking [50]. The relationship between literacy level and early healthcare seeking may also be because people who are literate are better able to access and understand health education messages from different media and health professionals [51]. Patients who were not members of the CBHI were also 7.14 times more likely to be delayed seeking healthcare than those who were CBHI members. Health insurance membership prevents patients from paying for laboratories and medications during a visit, which may be a reason that patients seek healthcare sooner. Furthermore, patients without a family history of death were 3 times more likely to be delayed than patients with a family history of death wre. This result was supported by studies conducted in China among imported malaria cases [52] and Nigeria [47]. This may indicate that patients with a family history of death recognize risk and severity of illness to a greater degree than patients without a history of family death. These findings suggest that perceptions of disease severity influence healthcare seeking behavior in the community.

Patients who went to health institutions on foot or by horse were 2.77 and 2.76 times more likely to be delayed than those who went by car, respectively. This is because patients may spend a long time traveling long distance on feet or by horse to health institutions. They may also delay seeking healthcare because they fear walking long distances or want to wait for their recovery to progress. Furthermore, patients who lived far from a health facility were 2.65 times more likely to be delayed than patients who came from near a health facility. This result was supported by studies conducted in Myanmar [41], Kenya [49] and Equatorial Guinea [53]. This could be due to the local advantage of being close to medical facilities. In addition, the results also indicate that transportation costs contribute to delays in seeking healthcare among malaria patients [54]. Access to health facilities near patients' villages can help increase awareness of early diagnosis and help patients seek timely healthcare [13,54,55].

## Conclusion

In this meta-analysis, the pooled prevalence of delays in seeking healthcare among malaria patients in Ethiopia was high. It also varies among different study settings and regions of the country. Furthermore, delays in seeking healthcare among malaria patients were affected by age, level of education, mode of transportation, distance to the health institution, family history of death and CBHI membership. The overall determinants of healthcare seeking delay in the current meta-analysis were children younger than 15 years, patients who could not read and write, patients who went to health institutions by foot or horse, patients who lived far from health institutions, patients who were not members of the CBHI and patients who had no family history of death. Hence, most of these determinants are modifiable. These findings underscore the need for targeted interventions to address these barriers and improve timely access to healthcare for affected populations.

## Recommendations

Health professionals and other stakeholders should focus on health education and mobilization of communities in malaria-endemic areas to seek healthcare early with a focus on major symptoms. Health education should provide a special focus on the children of parents and those who cannot read and write. The government should build infrastructure, particularly roads, to lessen the barrier of transportation in accessing healthcare facilities.

## Strengths and limitations

The systematic review and meta-analysis used an inclusive search that considered both published and unpublished studies. Study selection, data extraction and critical appraisal were conducted with more than one reviewer to prevent the possibility of bias. The analysis also included subgroup analysis and sensitivity analysis to handle heterogeneity among studies.

Patients and parents of children may not be able to clearly state the exact time of symptom onset, as 24 hours was set as the cutoff point, which may have resulted in recall bias in each primary study. We conducted subgroup analyses in a region with a single study, which may have affected our conclusions. The presence of high heterogeneity and publication bias among the included studies were also other limitations.

## Supporting information

**S1 File. This is PRISMA 2020 checklist.**
(DOC)

**S2 File. This is JBI Critical Appraisal checklist.**
(DOC)

**S3 File. This is extracted data for the included studies.**
(XLSX)

**S4 File. This is study identified through literature search.**
(XLSX)

## Acknowledgments

We would like to thank all the authors of the studies included in this systematic review and meta-analysis.

## Author contributions

**Data curation:** Dessie Alemnew Shiferaw, Tadele Kassahun Wudu.

**Formal analysis:** Moges Tadesse Abebe, Tesfahun Zemene Tafere, Kaleab Tesfaye Tegegne, Dessie Alemnew Shiferaw, Tadele Kassahun Wudu, Getnet Azanaw Takele.

**Funding acquisition:** Getnet Azanaw Takele.

**Investigation:** Moges Tadesse Abebe, Dessie Alemnew Shiferaw, Yosef Aragaw Gonete, Tadele Kassahun Wudu.

**Methodology:** Moges Tadesse Abebe, Tesfahun Zemene Tafere, Kaleab Tesfaye Tegegne, Dessie Alemnew Shiferaw, Yosef Aragaw Gonete.

**Project administration:** Moges Tadesse Abebe, Kaleab Tesfaye Tegegne, Dessie Alemnew Shiferaw, Getnet Azanaw Takele.

**Resources:** Moges Tadesse Abebe, Dessie Alemnew Shiferaw, Yosef Aragaw Gonete.

**Software:** Moges Tadesse Abebe, Dessie Alemnew Shiferaw, Tadele Kassahun Wudu, Getnet Azanaw Takele, Muluken Chanie Agimas.

**Supervision:** Moges Tadesse Abebe, Kaleab Tesfaye Tegegne, Yosef Aragaw Gonete, Tadele Kassahun Wudu, Getnet Azanaw Takele, Muluken Chanie Agimas.

**Validation:** Moges Tadesse Abebe, Dessie Alemnew Shiferaw, Yosef Aragaw Gonete, Muluken Chanie Agimas.

**Visualization:** Moges Tadesse Abebe, Kaleab Tesfaye Tegegne, Yosef Aragaw Gonete, Tadele Kassahun Wudu, Getnet Azanaw Takele, Muluken Chanie Agimas.

**Writing – original draft:** Moges Tadesse Abebe, Kaleab Tesfaye Tegegne, Dessie Alemnew Shiferaw, Yosef Aragaw Gonete, Tadele Kassahun Wudu, Getnet Azanaw Takele, Muluken Chanie Agimas.

**Writing – review & editing:** Moges Tadesse Abebe, Kaleab Tesfaye Tegegne, Dessie Alemnew Shiferaw, Yosef Aragaw Gonete, Tadele Kassahun Wudu, Muluken Chanie Agimas.

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
