## [Decision Letter · Decision Letter 0]

29 Dec 2024

PONE-D-24-28765Delay in healthcare seeking and its determinants among malaria patients in Ethiopia : a systematic review and meta-analysis.PLOS ONE

Dear Dr. Tadesse Abebe,

manuscript needs serious English edition by language professionalsIn abstract1. Identifying the pooled prevalence and determinants of delay ........ How to identify pooled prevalence? revise  the word identify.2 . avoid abbreviation in abstract3. the author used the term magnitude and prevalence alternatively. are the two words are the same? what is the objective of the study?. make consistent throughout the document.method1.A search of PROSPERO database was conducted and it was not register in it. what does it mean. rewrite it2. you stated as <Unpublished studies were searched from the online Universities institutional repositories>. did all universities in Ethiopia has repositories?3.The search terms were: “Malaria”, “patient delay”, “treatment seeking delay”, “healthcare seeking delay", and “Ethiopia”. The Boolean operators (AND and OR) were used to combine terms. How the Boolean operators (AND/OR) were applied to combine the search terms. Specifically, could you provide an example of the exact search string or logic you used when searching the databases?4.IN PRISMA Flow chart of study selection, 1171 records retrieved from combined searches an 242 records screened. by what exclusion criteria removed the rest records? give scientific reason for selection of sources?5. how many unpublished sources you included in your search

We look forward to receiving your revised manuscript.

Kind regards,

Bedilu Linger Endalifer

Academic Editor

PLOS ONE

https://pedsjazan.wordpress.com/wp-content/uploads/2013/09/eseential-nelson-ed-6.pdf

https://journals.plos.org/plosone/article?id=10.1371%2Fjournal.pone.0253746

In your revision ensure you cite all your sources (including your own works), and quote or rephrase any duplicated text outside the methods section. Further consideration is dependent on these concerns being addressed.

3. As required by our policy on Data Availability, please ensure your manuscript or supplementary information includes the following:

4. Please include a caption for figure 1, 3, 4, 5.

5. We note you have included a table to which you do not refer in the text of your manuscript. Please ensure that you refer to Table 2 in your text; if accepted, production will need this reference to link the reader to the Table.

Reviewers' comments:

Reviewer's Responses to Questions

**Comments to the Author**

1. Is the manuscript technically sound, and do the data support the conclusions?

Reviewer #1: Partly

Reviewer #2: Yes

Reviewer #3: Partly

Reviewer #4: Yes

Reviewer #5: Yes

2. Has the statistical analysis been performed appropriately and rigorously? 

Reviewer #1: Yes

Reviewer #2: Yes

Reviewer #3: Yes

Reviewer #4: Yes

Reviewer #5: Yes

3. Have the authors made all data underlying the findings in their manuscript fully available?

Reviewer #1: No

Reviewer #2: Yes

Reviewer #3: Yes

Reviewer #4: Yes

Reviewer #5: No

4. Is the manuscript presented in an intelligible fashion and written in standard English?

Reviewer #1: Yes

Reviewer #2: No

Reviewer #3: No

Reviewer #4: No

Reviewer #5: No

5. Review Comments to the Author

Reviewer #1: The manuscript presents an important health condition and its determinants. The authors did good effort to produce this manuscript. Providing some operational definitions was very good.

The study is conducted as a systematic review but it includes fundamental flaws that require major revision.

Considering the reporting guidelines of systematic reviews and its quality check tools, here are the major concerns:

-The reporting of the Abstract did not follow a guideline and many fundamental infos are missing from it. Please Check PRISMA-Abstract guidance.

- The methods did not report the screening process.

- there is no clear justification for using only the high quality studies

- the language restriction to English is not justified, especially that it is not the native tongue of the country addresses.

- A list of references excluded in the full text screening is missing

- items extracted per study not very sufficient to understand the results section. for example, age is found to be one of the determinants but the reader cannot figure out how many studies included children or adults... Also, sex is completely missed from the determinants and it could be important to elaborate about it.

- More details about the conducted meta-analysis are missing.

- Discussion does not need to start with a background paragraph. instead, it should start with a brief summary of your manuscript aim and, at least, the most important findings of your analysis.

Other minor concerns:

- the first sentence in the searching subheading related to Prospero is not clear!

- the sources of the grey literature is not clear.. do yo mean Ethiopian universities?

- the introduction need some improvement of the paragraphs length! Please keep it consistent and cohesive paragraphs.

- Cite the included studies in the first mention. i.e., in this sentence " this systematic review and meta-analysis included 18 articles.."

This manuscript has a good potential to be published and we look forward to reading your revised version

Reviewer #2: Abebe et al. present a comprehensive systematic review and meta-analysis of the pooled prevalence and determinants of delayed healthcare seeking behaviours amongst malaria patients in Ethiopia. The authors outline their rationale and study objectives, employ and describe a rigorous methodology, and clearly report their findings. The interpretation of results is well supported and the adhere to PRISMA reporting guidelines. Overall, my comments pertain to the expression of the results and the English language that could be clarified as per the suggestions below.

Comments:

1. Competing Interest Statement – doesn’t exactly match the example statement provided.

2. Overall: Manuscript should be proof-read for English language comprehension – with care taken to ensure correct tenses are used throughout the paper. However, the authors meaning is still intelligible. Some examples of common spelling mistakes: till=until, Hoarse=horse, etc.

3. Overall: The terms “Healthcare seeking delay”, “Treatment seeking delay”, “Patient Delay”, “Care-seeking delay” are used interchangeably, paper could be strengthened either by using consistent terminology or including definitions of these terms and how they differ from one another.

4. Abstract: should be amended to “patients who had NO family history of death”

5. Abstract: statement of “Delays in healthcare seeking among malaria patients were associated with an increased risk of severe disease and mortality. Identifying the pooled prevalence and determinants of delay in seeking healthcare in Ethiopia helps to reduce morbidity and mortality among malaria patients” is not something actually investigated in this systematic review and meta-analysis. Perhaps rephrase to: “Delays in healthcare seeking among malaria patients are typically associated with an increased risk of severe disease and mortality. Therefore, identifying the pooled prevalence and determinants of these delay in seeking healthcare in Ethiopia may help to reduce morbidity and mortality among malaria patients”

6. Abstract: suggest changing magnitude to prevalence

7. Introduction: would benefit for a short description of severe malaria and description of why early healthcare intervention is important in its prevention.

8. Introduction: This statement could be clarified “A meta-analysis study showed that a delay of treatment seeking >24 hours for uncomplicated Plasmodium falciparum infection was more likely to progress to severe complicated malaria (12).”

9. Introduction: Discussion of CMHI comes out of nowhere, maybe incorporate into the paragraph below or into you definitions?

10. Introduction: Suggest improving the flow of the final paragraph to be a clear statement of the objectives of the systematic review and meta-analysis and use this as Item #4 for PRISMA

11. Methods: was the systematic review registered with PROSPERO. If so, what was the number, etc. Update the PRISMA guidelines to reflect this

12. Results: Table 1 indicates that Tesfahunegn et al is a cross sectional study, so why hasn’t it been included in the pooled prevalence estimate?

13. Results: when discussing results, used pooled proportion/prevalence rather than effect size.

14. Results: The pooled prevalence estimates of the subgroup analysis are reported in the text as OR, when they should be pooled proportions/prevalences. Or the authors need to explain what the exposure/reference group is.

15. Results: If Dida, Darega et al. region is Southwest, why is it included as Oromia in the pooled analysis?

16. Figures 2-4: Make x-axis consistent (remove line at 0 for Fig 2). Improve labelling (ES should be pooled proportion and 95%CI). More informative headings could be used for Figs 3 and 4 respectively, e.g. “Pooled prevalence of delayed access to healthcare for malaria in Ethiopia, by region”.

17. Results: “Experience of death among family members was also determinants of delay” and “Being a member of a CBHI was a significant determinant of delay” – when actually they are determinants of early healthcare seeking behaviours.

18. Discussion: Overall, I found the discussion to be a bit repetitive, To improve, I would suggest discussing similar findings could be interpreted together (e.g. distance between residence and health-care settings AND the mode of transport to get there (i.e. foot/horse)). Similarly, age->education->CBHI->Family history of death could be discussed together.

19. Discussion: The OR provided for patients who travelled to health institutions on foot is 2.82, while in the results section it is 2.77

20. Discussion: I would suggest removing the statement of “This may be the reason why malaria morbidity and mortality is higher in children than in adults.” Children are more likely to become seriously ill/develop severe malaria due to a lack of naturally acquired immunity

Reviewer #3: I would like to thank the editors for the opportunity to review this paper and the authors for their thorough meta-analysis investigating delays in receipt of care for malaria and determinants of delay in Ethiopia. This a very clinically important topic as delays in malaria treatment can be fatal, especially in vulnerable populations such as children, as highlighted by the authors, and also there are important implications for the WHO malaria Test and Treat policy if delayed healthcare seeking delays both testing and treatment in malaria endemic settings. Overall, the methodology and conclusions from the data appear to be sound, but there are a few inconsistencies or lack of detail in data reporting and a lack of clarity in the writing in places. I have included detailed comments below for authors organized by section:

Abstract

1. Needs to include a clear statement about study objective (Item 2 of the PRISMA 2020 Abstract checklist).

2. Needs to include the inclusion criteria and exclusion criteria, including definition of delay that was used in methods section (Item 3 of the PRISMA 2020 Abstract checklist).

3. Recommend rewording the first sentence of the ‘Conclusion’ section. The findings of this study are not consistent with high magnitude of healthcare delay (i.e. a long delay prior), but a high prevalence of delay.

Introduction

1. Could be strengthened by providing more context in terms of WHO malaria test and treat policy.

2. Referencing needs to be more precise. For example, the 4th paragraph of the introduction session includes the phrase ‘studies conducted in several regions of Ethiopia found that a relatively few patients actually seek medical care within the recommended 24 hours after the onset of symptoms’, but there are no references in this sentence to the studies mentioned. Likewise, paragraph 6 begins ‘Among studies conducted in Ethiopia, several factors influences delays in seeking malaria treatment. These include socio-demographic factors such as age, gender, marital status, educational status, occupational status and income’, however, no references are provided at this point.

3. The final sentence of the introduction needs to more clearly written- it needs to more clearly state that it is the prevalence and determinants of delayed healthcare seeking that were the focus of the current study and that they were conducting the first systematic review in Ethiopia (as opposed to an expert review of the literature available).

Methods

1. Under ‘Searching’ subheading, the authors do not state which University repositories they searched. Was this their own institution? Or multiple institutions in Ethiopia?

2. After reading the definition and inclusion/exclusion criteria, it is still not entirely clear what definition was used for the cutoff for delayed care. The inclusion criteria says studies using 24 hours as a cutoff were included, but exclusion criteria state studies using a cutoff of >48 hours were excluded. What was done if the studies used a cutoff between 24 and 48 hours? Were these included or excluded?

3. ‘The outcome of the study’ subsection should be more specific in stating the outcome was the prevalence of delay.

4. The ‘study selection’ subsection and PRISMA flow chart (figure 1) do not contain sufficient information about why studies were excluded. More information should be included about the 7 articles that were excluded for ‘data inconsistency’ as I am not sure what the authors mean by this. Figure 1 needs additional data for clarity about the studies not included in the final analysis, including the specific reasons for the 116 studies being ‘excluded with reason’ and the 108 studies in the ‘Tittle and abstract not met’ section (and I suggest spelling of title be corrected).

Results

1. There are some key inconsistencies in the results section in reporting which studies were used in the key outcome of determining prevalence of delay. In the ‘prevalence of delay’ subheading in the results section, it states only 4815 participants were used in the analysis from ten cross-sectional studies. I understand that case-control studies could not be used to estimate prevalence, however, there are only nine cross-sectional and nine case-control studies listed in the JBI critical appraisal tables (S2) and Table 1 lists 11 cross-sectional studies and 7 case control studies. These discrepancies need to be rectified in the results text and Table 1 and supplemental table. It is unclear whether 9, 10 or 11 cross-sectional studies were included in this part of the metaanalysis.

2. In the ‘Subgroup analysis’ subheading, the authors reference several times the phrase ‘more delayed’ when referring to Odd ratios of delayed care e.g. ‘children less than 15 years were 2.27 times more delayed than adults’. This use of ‘more delayed’ leads the reader to misinterpret the result as saying that the magnitude of delay was 2.27 times greater in children, when in fact I think the authors mean that children less than 15 years were 2.27 times more likely to be delayed. I recommend the authors review their subgroup analysis section and replace the incorrect use of ‘more delayed’ throughout this section.

3. In the ‘Subgroup analysis’ subheading, in the last paragraph of page 16 for example, the authors refer to ‘experience of death among family members’ and patients with ‘no family history of death’. I am not sure what this means exactly, or how it was defined. Table 2 also includes the phrase ‘had no history of death’ as a determinant and I am unclear what this refers to. Please clarify whether the authors are referring to recent experience of death of a family member in these sections and if so, how this was defined?

Discussion

1. Paragraph 2 of the discussion refers to ‘percentage of delayed healthcare seeking’ in the first sentence. I suggest the authors change this to prevalence of delayed healthcare seeking for clarity and readability and to be consistent with other parts of the manuscript.

2. Final paragraph of the discussion also references ‘family history of death’ as a determinant of delayed healthcare seeking, but it is still how this was defined.

3. Discussion could be strengthened by considering whether regional differences and the one study noted to have a large effect on the primary outcome, could be explained by differences in rural versus urban populations, or lower versus higher income population settings for individual studies, and whether these factors may represent unmeasured confounders.

4. There is a misspelling of horse (the animal) as ‘hoarse’ in the last paragraph of page 20 in the discussion section. Please correct.

Tables/Figures

1. Tables are generally clearly presented but headings need to use correct capitalization.

Reviewer #4: General comments:

• Overall, an important meta-analysis for malaria care in Ethiopia but improvements are required.

• Authors indicate that data is available but not where it is available from. Suggest copying example statement to indicate that all data is available in the manuscript or supplementary material

• Have indicated some instances were English/grammar should be revised through the beginning of the methods but overall, the manuscript would benefit from review for English and grammatical correctness, as well as from copyediting.

• Consistently write healthcare or health care.

• Check Italicizations.

• Check tense used.

• The introduction lists all these determinants for delays in care seeking. I think the rationale for this meta-analysis needs to be more clearly articulated.

• Google (vs Google Scholar) is not usually used as a search database for systematic reviews. What was the justification of including it as one?

• Flow of the discussion could be improved.

Abstract:

• Methods section of abstract suggest consistently capitalization the names of databases searched.

• Last sentence of the methods would add “a” before “individual study”

• In the results, would suggest presenting the confidence interval as a range as opposed to using a comma between the start and end of the range.

• With a p-value of 0.002 for the Egger test would this suggest that that it is highly significant not just significant?

• In the last sentence of the results section, you list the determinants of delay in seeking healthcare among malaria patients in Ethiopia. I would suggest omitting patients before each determinant. For instance instead of writing, “patients less than 15 years old” I would write, “being under 15 years old, not being able to read and write, traveling to the health center on foot,…, were significant determinants of delay……” because the patients themselves are not the determinants.

• In the conclusion of the abstract, it summarizes what was already presented in the results. Are there any broader conclusions or suggestions that can be made based on these findings?

Introduction:

• Consistent italicizing of full malaria species name (including the genus)

• Page 4: Suggest changing responsible to cause…. “To responsible for most cases of severe malaria.” Though less common, other species can also cause severe malaria.

• Page 4: “Global malaria prevalence increased.....” Change increases to increased.

• Page 4: Do not need the “And” at the start of the last sentence of the second paragraph.

• Page 4-5: “Depending on the nature of the epidemic in each district..” change to “Depending on transmission dynamics in each district….

• Page 5: it is unclear what the meta-analysis indicates mentioned in the second sentence of the second paragraph. That is not treated within 24 hours that it would progress to severe malaria? If so, would clarify this.

Page 5: “According to the recent Ethiopian and WHO national malaria control programs,…” recent what? And the WHO does not have a national program, so according to recent evidence, the WHO and Ethiopian national malaria control program,…..”

• Page 5, paragraph 3: CBHI acronym, missing the B in the full written name? Community-Based Health Insurance?

• Page 6: would explain what “chat” is for those who may not know.

• Page 6: “There is also inconsistent finding among…” -> “There are also inconsistent findings among…”

• Page 6: Last sentence of the first paragraph needs clarifying. It does not make sense. Seems both findings are the same. Tense use is also incorrect.

Methods:

• Page 7, under searching: in the first sentence, would clarify what was not registered in PROSPERO.

• Page 7: Consistent capitalization of databases searched.

A search was understand taken, not were.

• Page 7: what universities’ institutional repositories?

• Page 8: All inclusion criteria after the first one listed, just appear to be more specifics about the first inclusion criteria. If so, should they be inclusion criteria? They fit better as listed in the exclusion criteria.

Results:

• Page 14: Prevalence of delay: suggest to maybe clarify why those 10 cross-sectional studies were included in the meta-analysis (as opposed those that were not)?

• Page 14: add a comma in the first sentence after meta-analysis

• Page 14: here results are presented from a subgroup analysis. I would suggest adding more detail about this in the methods section. Also is this subgroup analysis methodologically sound since some regions had a lot fewer studies conducted than others?

• Page 15-16: I would suggested in the last sentence of the last paragraph to change it to “Of these patient-related variables, seven of them (being less than 15 years old, xxxxx),… were significantly associated with delay…” The patients themselves are not the variables.

• Page 16: “Age was the determinant of delay in seeking health care among malaria patients.” Here to you mean that age has the most impact? For all the different factors you talk about, I would make a statement as to how their impact relates to one another.

• Page 16: Misspelling of horse in the second paragraph

Discussion:

• Page 18: “This result was supported by studies conducted in Myanmar(42) and Tanzania (43), but it was lower than study conducted in China in which 85.6% where of patients were delayed (41).” Would specify what result you mean here, since the sentence before you are talking about the recommendation.

• Overall, the flow of the discussion is okay, though within some paragraphs the flow is a bit disjointed with findings, other studies, and recommendations not discussed at once and broken apart.

• Suggest having a stronger strength and limitations section/paragraph at the end of the discussion.

Reviewer #5: Review

The manuscript “Delay in healthcare seeking and its determinants among malaria patients in Ethiopia: a systematic review and meta-analysis” in my opinion requires major revision. The manuscript is difficult to read in several areas and some points are unclear due to “word order” and the language used. Throughout the article, English should be proofread. There are, however, some significant comments that need to be addressed.

Abstract

First, the abstract needs some revisions. The methods must be an accurate reflection of what was done as per the Methods section. The results can be summarised.

2. Introduction

The authors should provide references, when factual statements are made, at the end of the sentence rather than at the end of the paragraph. Paragraphing should be improved and single sentence paragraphs should be avoided.

3. Methods

Google Web has been shown to consistently have the lowest quality relevance ratio and contain many duplicate hits when it comes scientific literature searches. I find it surprising the authors used Google Scholar and Google as well. Authors should provide the MESH terms used and indicate whether the same search strategy was used for all the databases. A repetition of part of the exclusion criteria appears under the Data quality control measures. The sub-heading “Study selection” should be part of the Results.

4. Results

In Table 1 authors should indicate the reference numbers of the studies included in their work. Dejazach, Alemu et al. cannot be found under the Reference section. Several sections of the results need to be rewritten for clarity. Figures have not been numbered making identification difficult. No table 3 seen making it impossible to determine why the 7 factors were chosen.

5. Discussion

Discussion can be markedly improved using recent publications. Some factual statements have been made without any supporting citations. Some sections also need to be rewritten for clarity. The limitations should be expanded.

6. Conclusion

Conclusion has repetitions and this should be addressed.

References

Please review your reference list to ensure that it is complete and correct.

6. PLOS authors have the option to publish the peer review history of their article (what does this mean? ). If published, this will include your full peer review and any attached files.

**Do you want your identity to be public for this peer review?** For information about this choice, including consent withdrawal, please see our Privacy Policy .

Reviewer #1: No

Reviewer #2: No

Reviewer #3: No

Reviewer #4: No

Reviewer #5: No

---

## [Author Response · Author response to Decision Letter 1]

17 Jan 2025

Response to the editor and reviewers

Editor’s comments

1. manuscript needs serious English edition by language professionals.

Response: Thank you. We have edited our manuscript accordingly.

Abstract

2. Identifying the pooled prevalence and determinants of delay ........ How to identify pooled prevalence? revise the word identify.

Response: thank you. We have replaced the word “identifying” with “determining” (Page 2, line 3)

3. Avoid abbreviation in abstract

Response: thank you. We have avoided abbreviation in abstract.

4. the author used the term magnitude and prevalence alternatively. are the two words are the same? what is the objective of the study?. make consistent throughout the document.

Response: thank you. We changed the word “magnitude” with “prevalence” thought the document. The objective of this systematic review and meta-analysis was to determine the pooled prevalence of delay in seeking healthcare and its determinants among malaria patients in Ethiopia (page 2, line 4-7).

Method

1. A search of PROSPERO database was conducted and it was not register in it. what does it mean. rewrite it

Response: thank you. We intended to say that we have checked any related title registered in the PROSPERO database to prevent duplicated work, and there was no similar registered review in the PROSPERO database (page 7 line 6-8).

2. you stated as <Unpublished studies were searched from the online Universities institutional repositories>. did all universities in Ethiopia has repositories?

Response: thank you. Addis Ababa and Jima universities have online repositories and we have checked them. In addition, we conducted Google and Google Scholar search to include all available gray and published literatures conducted in Ethiopia, although we didn’t find any unpublished article online (page 7).

3. The search terms were: “Malaria”, “patient delay”, “treatment seeking delay”, “healthcare seeking delay", and “Ethiopia”. The Boolean operators (AND and OR) were used to combine terms. How the Boolean operators (AND/OR) were applied to combine the search terms. Specifically, could you provide an example of the exact search string or logic you used when searching the databases?

Response: thank you. PubMed search string: ((Malaria[MeSH Terms]) AND (Patient delay OR treatment seeking delay OR seeking healthcare delay[MeSH Terms])) AND (Ethiopia) were searched. (page 7).

4. In PRISMA Flow chart of study selection, 1171 records retrieved from combined searches and 242 records screened. by what exclusion criteria removed the rest records? give scientific reason for selection of sources?

Response: thank you. We have written the full exclusion criteria on selection results section (page 11), and PRISMA flow chart (Fig 1).

5. how many unpublished sources you included in your search

Response: thank you. For comprehensive searching, we tried to search for unpublished articles conducted in Ethiopia. However, we didn’t find any unpublished article to be included in our review.

Additionally,

1. We have written the manuscript based on PLOS ONE`s style requirements.

2. We have removed the occurrence of overlapping texts and cited all sources.

3. We have included a numbered table of literature search, including those that were excluded including the reasons for exclusion, name of data extractor, eligibility and date of search (S4 Table).

4. Quality assessment table (S2 Table).

5. The explanation for how missing data were handled (page 11)

6. We have included a caption for all figures.

7. We have referred all tables in the text.

Reviewer 1

Reviewer #1: The manuscript presents an important health condition and its determinants. The authors did good effort to produce this manuscript. Providing some operational definitions was very good.

The study is conducted as a systematic review but it includes fundamental flaws that require major revision.

Considering the reporting guidelines of systematic reviews and its quality check tools, here are the major concerns:

-The reporting of the Abstract did not follow a guideline and many fundamental infos are missing from it. Please Check PRISMA-Abstract guidance.

Response: thank you. We have checked the PRISMA-Abstract guidance and re-written accordingly (page 2).

- The methods did not report the screening process.

Response: thank you. We have reported the screening process at the methods section (page 9).

- there is no clear justification for using only the high quality studies

Response: thank you. To the best of our knowledge, we have appraised included articles, and the included articles fulfilled the inclusion criteria of critical appraisal checklists.

- the language restriction to English is not justified, especially that it is not the native tongue of the country addresses.

Response: thank you. The language restriction to English was because databases retrieved articles conducted with other languages during the selection process.

- A list of references excluded in the full text screening is missing

Response: thank you. Since large numbers of articles were excluded after full text reading, we have cited 18 included articles (page 11).

- items extracted per study not very sufficient to understand the results section. for example, age is found to be one of the determinants but the reader cannot figure out how many studies included children or adults... Also, sex is completely missed from the determinants and it could be important to elaborate about it.

Response: thank you. We have included the number of studies conducted among children and adults in the results section of characteristics of the study (page 11). Concerning to sex, we have extracted and analyzed, but it was found to be insignificant. (See S3 Table)

- More details about the conducted meta-analysis are missing.

Response: thank you. We have added the conducted meta-analysis for pooled prevalence and determinants (page 13&15)

- Discussion does not need to start with a background paragraph. instead, it should start with a brief summary of your manuscript aim and, at least, the most important findings of your analysis.

Response: thank you. We have corrected the first paragraph of our discussion accordingly (page 18).

Other minor concerns:

- the first sentence in the searching subheading related to Prospero is not clear!

Response: thank you. We have corrected to make it clear (page 7)

- the sources of the grey literature is not clear. do yo mean Ethiopian universities?

Response: thank you. Yes, the online universities repository was searched to check availability of grey literatures, but we didn’t find any gray literature related to our title. Additionally, Google and Google Scholar search was used as a source of both gray and published literatures from other sources out of the included databases.

- the introduction need some improvement of the paragraphs length! Please keep it consistent and cohesive paragraphs.

Response: thank you. We have revised our introduction.

- Cite the included studies in the first mention. i.e., in this sentence " this systematic review and meta-analysis included 18 articles.."

Response: thank you. We have cited the included articles (page 11)

This manuscript has a good potential to be published and we look forward to reading your revised version

Response: thank very much.

Reviewer #2: Abebe et al. present a comprehensive systematic review and meta-analysis of the pooled prevalence and determinants of delayed healthcare seeking behaviours amongst malaria patients in Ethiopia. The authors outline their rationale and study objectives, employ and describe a rigorous methodology, and clearly report their findings. The interpretation of results is well supported and the adhere to PRISMA reporting guidelines. Overall, my comments pertain to the expression of the results and the English language that could be clarified as per the suggestions below.

Comments:

1. Competing Interest Statement – doesn’t exactly match the example statement provided.

Response: thank you. We have corrected the competing interest statement.

2. Overall: Manuscript should be proof-read for English language comprehension – with care taken to ensure correct tenses are used throughout the paper. However, the authors meaning is still intelligible. Some examples of common spelling mistakes: till=until, Hoarse=horse, etc.

Response: thank you. We have conducted proof-read for English language comprehension.

3. Overall: The terms “Healthcare seeking delay”, “Treatment seeking delay”, “Patient Delay”, “Care-seeking delay” are used interchangeably, paper could be strengthened either by using consistent terminology or including definitions of these terms and how they differ from one another.

Response: thank you. We have used a phrase “delay in seeking healthcare” consistently.

4. Abstract: should be amended to “patients who had NO family history of death”

Response: thank you. We have corrected accordingly (page 3).

5. Abstract: statement of “Delays in healthcare seeking among malaria patients were associated with an increased risk of severe disease and mortality. Identifying the pooled prevalence and determinants of delay in seeking healthcare in Ethiopia helps to reduce morbidity and mortality among malaria patients” is not something actually investigated in this systematic review and meta-analysis. Perhaps rephrase to: “Delays in healthcare seeking among malaria patients are typically associated with an increased risk of severe disease and mortality. Therefore, identifying the pooled prevalence and determinants of these delay in seeking healthcare in Ethiopia may help to reduce morbidity and mortality among malaria patients”

Response: thank you. We have corrected accordingly (page 2).

6. Abstract: suggest changing magnitude to prevalence

Response: thank you. We have changed magnitude with prevalence (page 2).

7. Introduction: would benefit for a short description of severe malaria and description of why early healthcare intervention is important in its prevention.

Response: thank you. We have added about severe malaria and the description of why early health intervention is important (page 4&5).

8. Introduction: This statement could be clarified “A meta-analysis study showed that a delay of treatment seeking >24 hours for uncomplicated Plasmodium falciparum infection was more likely to progress to severe complicated malaria (12).”

Response: thank you. We have clarified the sentences (page 4&5).

9. Introduction: Discussion of CMHI comes out of nowhere, maybe incorporate into the paragraph below or into your definitions?

Response: thank you. We have removed the paragraph.

10. Introduction: Suggest improving the flow of the final paragraph to be a clear statement of the objectives of the systematic review and meta-analysis and use this as Item #4 for PRISMA

Response: thank you. We have corrected accordingly (page 6).

11. Methods: was the systematic review registered with PROSPERO. If so, what was the number, etc. Update the PRISMA guidelines to reflect this

Response: thank you. This systematic was not registered with PROSPERO. We have updated the PRISMA guideline

12. Results: Table 1 indicates that Tesfahunegn et al is a cross-sectional study, so why hasn’t it been included in the pooled prevalence estimate?

Response: thank you. Tesfahunegn et al is a case control study. We have corrected it (page 12).

13. Results: when discussing results, used pooled proportion/prevalence rather than effect size.

Response: thank you. We have corrected accordingly.

14. Results: The pooled prevalence estimates of the subgroup analysis are reported in the text as OR, when they should be pooled proportions/prevalences. Or the authors need to explain what the exposure/reference group is.

Response: thank you for reminding us this concept. It is proportion not OR. We have corrected it accordingly (page 13).

15. Results: If Dida, Darega et al. region is Southwest, why is it included as Oromia in the pooled analysis?

Response: thank you. We have corrected it as Oromia (page 12).

16. Figures 2-4: Make x-axis consistent (remove line at 0 for Fig 2). Improve labelling (ES should be pooled proportion and 95%CI). More informative headings could be used for Figs 3 and 4 respectively, e.g. “Pooled prevalence of delayed access to healthcare for malaria in Ethiopia, by region”.

Response: thank you. The line at 0 is automatic STATA line that we could not manipulate to remove it. We have improved the headings accordingly.

17. Results: “Experience of death among family members was also determinants of delay” and “Being a member of a CBHI was a significant determinant of delay” – when actually they are determinants of early healthcare seeking behaviours.

Response: thank you. We have corrected. Experience of no death among family members was also determinants of delay. Not being a member of a CBHI was a significant determinant of delay (page 16).

18. Discussion: Overall, I found the discussion to be a bit repetitive, To improve, I would suggest discussing similar findings could be interpreted together (e.g. distance between residence and health-care settings AND the mode of transport to get there (i.e. foot/horse)). Similarly, age->education->CBHI->Family history of death could be discussed together.

Response: thank you. We have written the discussion accordingly (page 18-22).

19. Discussion: The OR provided for patients who travelled to health institutions on foot is 2.82, while in the results section it is 2.77

Response: thank you. We have corrected it 2.77 (page 21).

20. Discussion: I would suggest removing the statement of “This may be the reason why malaria morbidity and mortality is higher in children than in adults.” Children are more likely to become seriously ill/develop severe malaria due to a lack of naturally acquired immunity

Response: thank you. We have removed it.

Reviewer #3: I would like to thank the editors for the opportunity to review this paper and the authors for their thorough meta-analysis investigating delays in receipt of care for malaria and determinants of delay in Ethiopia. This a very clinically important topic as delays in malaria treatment can be fatal, especially in vulnerable populations such as children, as highlighted by the authors, and also there are important implications for the WHO malaria Test and Treat policy if delayed healthcare seeking delays both testing and treatment in malaria endemic settings. Overall, the methodology and conclusions from the data appear to be sound, but there are a few inconsistencies or lack of detail in data reporting and a lack of clarity in the writing in places. I have included detailed comments below for authors organized by section:

Abstract

1. Needs to include a clear statement about study objective (Item 2 of the PRISMA 2020 Abstract checklist).

Response: thank you. We have added the objective to the abstract (page 2).

2. Needs to include the inclusion criteria and exclusion criteria, including definition of delay that was used in methods section (Item 3 of the PRISMA 2020 Abstract checklist).

Response: thank you. We have added the inclusion, exclusion and definition to the abstract (page 2).

3. Recommend rewording the first sentence of the ‘Conclusion’ section. The findings of this study are not consistent with high magnitude of healthcare delay (i.e. a long delay prior), but a high prevalence of delay.

Response: thank you. We have corrected accordingly (page 3).

Introduction

1. Could be strengthened by providing more context in terms of WHO malaria test and treat policy.

Response: thank you. We have strengthened the introduction (page 4-5).

2. Referencing needs to be more precise. For example, the 4th paragraph of the introduction session includes the phrase ‘studies conducted in several regions of Ethiopia found that a relatively few patients actually seek medical care within the recommended 24 hours after the onset of symptoms’, but there are no references in this sentence to the studies mentioned. Likewise, paragraph 6 begins ‘Among studies conducted in Ethiopia, several factors influences delays in seeking malaria treatment. These include socio-demographic factors such as age, gender, marital status, educational status, occupational s

---

## [Editor Report · Decision Letter 1]

14 Feb 2025

Delays in seeking healthcare and its determinants among malaria patients in Ethiopia : a systematic review and meta-analysis.

PONE-D-24-28765R1

Dear Dr.Moges Tadesse 

We’re pleased to inform you that your manuscript has been judged scientifically suitable for publication and will be formally accepted for publication once it meets all outstanding technical requirements.

Kind regards,

Bedilu Linger Endalifer

Academic Editor

PLOS ONE
---

## [Editor Report · Acceptance letter]

PONE-D-24-28765R1

PLOS ONE

Dear Dr. Tadesse Abebe,

I'm pleased to inform you that your manuscript has been deemed suitable for publication in PLOS ONE. Congratulations! Your manuscript is now being handed over to our production team.

Kind regards,

on behalf of

Dr. Bedilu Linger Endalifer

Academic Editor

PLOS ONE